

# Influence of temperature on the molecular composition of ions and charged clusters during pure biogenic nucleation

Carla Frege[1], Ismael K. Ortega[2], Matti P. Rissanen[3], Arnaud P. Praplan[3], Gerhard Steiner[3,4,5], Martin Heinritzi[6], Lauri Ahonen[3], António Amorim[7], Anne-Kathrin Bernhammer[4], Federico Bianchi[1,3], Sophia Brilke[4], Martin Breitenlechner[4], Lubna Dada[3], António Dias[7], Jonathan Duplissy[3,8], Sebastian Ehrhart[8], Imad El-Haddad[1], Lukas Fischer[4], Claudia Fuchs[1], Olga Garmash[3], Marc Gonin[9], Armin Hansel[4], Christopher R. Hoyle[1], Tuija Jokinen[3], Heikki Junninen[3], Jasper Kirkby[6,8], Andreas Kürten[6], Katrianne Lehtipalo[1,3], Markus Leiminger[6], Roy Lee Mauldin[3,16], Ugo Molteni[1], Leonid Nichman[10], Tuukka Petäjä[3], Nina Sarnela[3], Siegfried Schobesberger[3], Mario Simon[6], Mikko Sipilä[3], Dominik Stolzenburg[5], António Tomé[11], Alexander L. Vogel[1,8], Andrea Wagner[6], Robert Wagner[3], Mao Xiao[1], Chao Yan[3], Penglin Ye[12,15], Joachim Curtius[4], Neil M. Donahue[12], Rick C. Flagan[13], Markku Kulmala[3], Douglas R. Worsnop[3,14,15], Paul M. Winkler[5], Josef Dommen[1] and Urs Baltensperger[1]

[1] Paul Scherrer Institute, Laboratory of Atmospheric Chemistry, CH-5232 Villigen, Switzerland
[2] Onera -The French Aerospace Lab, F-91123 Palaiseau, France
[3] University of Helsinki, Department of Physics, P.O.Box 64, 00014 University of Helsinki, Finland
[4] University of Innsbruck, Institute of Ion Physics and Applied Physics, Technikerstraße 25, 6020 Innsbruck, Austria
[5] University of Vienna, Faculty of Physics, Boltzmanngasse 5, 1090 Vienna, Austria
[6] Institute for Atmospheric and Environmental Sciences, Goethe University Frankfurt, 60438 Frankfurt am Main, Germany
[7] Universidade de Lisboa, Ed. C8, Campo Grande, Lisboa- 1749-016, Portugal
[8] CERN, CH-1211 Geneva, Switzerland
[9] Tofwerk AG, 3600 Thun, Switzerland
[10] School of Earth and Environmental Sciences, University of Manchester, Manchester M13 9PL, UK
[11] IDL -Universidade da Beira Interior, Av. Marquês D'Avila e Bolama, 6201-001 Covilhã, Portugal.
[12] Center for Atmospheric Particle Studies, Carnegie Mellon University, Pittsburgh, Pennsylvania 15213, USA



[13] Division of Chemistry and Chemical Engineering, California Institute of Technology,
Pasadena, California 91125, USA
[14] University of Eastern Finland, FI-70211 Kuopio, Finland
[15] Aerodyne Research Inc., Billerica, Massachusetts 01821, USA
[16] Department of Atmospheric and Oceanic Sciences, University of Colorado, Boulder,
Colorado 80309-0311, USA.



## Abstract

It was recently shown by the CERN CLOUD experiment that biogenic highly oxygenated
molecules (HOMs) form particles under atmospheric conditions in the absence of sulfuric
acid where ions enhance the nucleation rate by one to two orders of magnitude. The biogenic
HOMs were produced from ozonolysis of α-pinene at 5°C. Here we extend this study to
compare the molecular composition of positive and negative HOM clusters measured with
atmospheric pressure interface time-of-flight mass spectrometers (APi-TOFs), at three
different temperatures (25°C, 5°C and -25°C). Most negative HOM clusters include a nitrate
($NO_3^-$) ion and the spectra are similar to those seen in the nighttime boreal forest. On the
other hand, most positive HOM clusters include an ammonium ($NH_4^+$) ion and the spectra are
characterized by mass bands that differ in their molecular weight by ~20 C atoms,
corresponding to HOM dimers. At lower temperatures the average oxygen to carbon (O:C)
ratio of the HOM clusters decreases for both polarities, reflecting an overall reduction of
HOM formation with decreasing temperature. This indicates a decrease in the rate of
autoxidation with temperature due to a rather high activation energy as has previously been
determined by quantum chemical calculations. Furthermore, at the lowest temperature
(-25°C) the presence of $C_{30}$ clusters show that HOM monomers start to contribute to the
nucleation of positive clusters. These experimental findings are supported by quantum
chemical calculations of the binding energies of representative neutral and charged clusters.

## 1. Introduction

Atmospheric aerosol particles directly affect climate by influencing the transfer of radiant
energy through the atmosphere (Boucher et al., 2013). In addition, aerosol particles can


indirectly affect climate, by serving as cloud condensation nuclei (CCN) and ice nuclei (IN).
They are of natural or anthropogenic origin, and result from direct emissions (primary
particles) or from oxidation of gaseous precursors (secondary particles). Understanding
particle formation processes in the atmosphere is important since more than half of the
atmospheric aerosol particles may originate from nucleation (Dunne et al., 2016; Merikanto
et al., 2009).
Due to its widespread presence and low saturation vapor pressure, sulfuric acid is believed to
be the main vapor responsible for new particle formation (NPF) in the atmosphere. Indeed,
particle nucleation is dependent on its concentration, albeit with large variability (Kulmala et
al., 2004). The combination of sulfuric acid with ammonia and amines increases nucleation
rates due to a higher stability of the initial clusters (Almeida et al., 2013; Kirkby et al., 2011;
Kürten et al., 2016). However, these clusters alone cannot explain the particle formation rates
observed in the atmosphere. Nucleation rates are greatly enhanced when oxidized organics
are present  together with sulfuric acid, resulting in NPF rates that closely match those
observed in the atmosphere (Metzger et al., 2010; Riccobono et al., 2014). An important
characteristic of the organic molecules participating in nucleation is their high oxygen content
and consequently low vapor pressure. The formation of these highly oxygenated molecules
(HOMs) has been described by Ehn et al. (2014), who found that, following the well-known
initial steps of α-pinene ozonolysis through a Criegee intermediate leading to the formation of
an $RO_2 \cdot$ radical, several repeated cycles of intramolecular hydrogen abstractions and $O_2$
additions produce progressively more oxygenated $RO_2 \cdot$ radicals, a mechanism called
autoxidation (Crounse et al., 2013). The (extremely) low volatility of the HOMs results in
efficient NPF and growth, even in the absence of sulfuric acid (Kirkby et al., 2016; Tröstl et
al., 2016). The chemical composition of HOMs during NPF has been identified from α-
pinene and pinanediol oxidation by Praplan et al. (2015) and Schobesberger et al. (2013),
respectively.
Charge has also been shown to enhance  nucleation (Kirkby et al., 2011). Ions are produced
in the atmosphere mainly by galactic cosmic rays and radon. The primary ions are $N^+$, $N_2^+$,
$O^+$, $O_2^+$, $H_3O^+$, $O^-$ and $O_2^-$ (Shuman et al., 2015). These generally form clusters with water
(e.g. $(H_2O)H_3O^+$) and after further collisions the positive and negative charges are transferred
to trace species with highest and lowest proton affinities, respectively (Ehn et al., 2010). Ions



are expected to promote NPF by increasing the cluster binding energy and reducing
evaporation rates (Hirsikko et al., 2011). Recent laboratory experiments showed that ions
increase the nucleation rates of HOMs from the oxidation of α-pinene by one to two orders of
magnitude compared to neutral conditions (Kirkby et al. 2016). This is due to two effects, of
which the first is more important: 1) an increase in cluster binding energy, which decreases
evaporation and 2) an enhanced collision probability, which increases the condensation of
polar vapors on the charged clusters (Lehtipalo et al., 2016; Nadykto, 2003).
Temperature plays an important role in nucleation, resulting in strong variations of NPF
at different altitudes. Kürten et al. (2016) studied the effect of temperature on nucleation for
the sulfuric acid - ammonia system, finding that low temperatures decrease the needed
concentration of $H_2SO_4$ to maintain a certain nucleation rate. Similar results have been found
for sulfuric acid – water binary nucleation (Duplissy et al., 2016; Merikanto et al., 2016),
where temperatures below 0˚C were needed for NPF to occur at atmospheric concentrations.
Up to now, no studies have addressed the temperature effect on NPF driven by HOMs from
biogenic precursors such as α-pinene.
In this study we focus on the chemical characterization of the ions and the influence of
temperature on their chemical composition during organic nucleation in the absence of
sulfuric acid. The importance of such sulfuric acid-free clusters for NPF has been shown in
the laboratory (Kirkby et al., 2016; Tröstl et al., 2016) as well as in the field (Bianchi et al.,
2016). We present measurements of the NPF process from the detection of primary ions (e.g.
$N_2^+$, $O_2^+$, $NO^+$) to the formation of clusters in the size range of small particles, all under
atmospherically relevant conditions. The experiments were conducted at three different
temperatures (-25, 5 and 25 ˚C) enabling the simulation of pure biogenic NPF representative
of different tropospheric altitudes.

## 2. Methods

### 2.1. The CLOUD chamber

We conducted experiments at the CERN CLOUD chamber (Cosmics Leaving Outdoor
Droplets). With a volume of 26.1 m³, the chamber is built of electropolished stainless steel





and equipped with a precisely controlled gas system. The temperature inside the chamber is
measured with a string of six thermocouples (TC, type K) which were mounted horizontally
between the chamber wall and the center of the chamber at distances of 100, 170, 270, 400,
650, and 950 mm from the chamber wall (Hoyle et al., 2016). The temperature is controlled
accurately (with a precision of $\pm$ 0.1°C) at any tropospheric temperature between -65 and
30 °C (in addition, the temperature can be raised to 100 °C for cleaning). The chamber
enables atmospheric simulations under highly stable experimental conditions with low
particle wall loss and low contamination levels (more details of the CLOUD chamber can be
found in Kirkby et al. (2011) and Duplissy et al. (2016)). Before the start of the experiments
the CLOUD chamber was cleaned by rinsing the walls with ultra-pure water, followed by
heating to 100°C and flushing at a high rate with humidified synthetic air and elevated ozone
(several ppmv) (Kirkby et al., 2016). This resulted in $SO_2$ and $H_2SO_4$ concentrations that
were below the detection limit (<15 pptv and $<5 \times 10^4$ cm$^{-3}$, respectively), and total organics
(largely comprising high volatility $C_1$–$C_3$ compounds) that were below 150 pptv.

The air in the chamber is ionized by galactic cosmic rays (GCR); higher ion generation

rates can be induced by a pion beam ($\pi^+$) from the CERN Proton Synchrotron enabling
controlled simulation of galactic cosmic rays throughout the troposphere. Therefore, the total
ion-pair production rate in the chamber is between 2 (no beam) and 100 cm$^{-3}$ s$^{-1}$ (maximum
available beam intensity, Franchin et al., 2015).

2.2. **Instrumentation**

The main instruments employed for this study were atmospheric pressure interface time-

of-flight (APi-TOF, Aerodyne Research Inc. & Tofwerk AG) mass spectrometers. The APi-
TOF is able to measure the intensity of positive or negative ions and cluster ions over a wide
range of mass-to-charge ratios at concentrations relevant for the ambient atmosphere. The
instrument has two main parts. The first is the atmospheric pressure interface (APi) where
ions are transferred from atmospheric pressure to low pressures via three differentially
pumped vacuum stages. Ions are focused and guided by two quadrupoles and ion lenses. The
second is the time-of-flight mass analyzer (TOF), where the pressure is approximately 10$^{-6}$
mbar. The sample flow from the chamber was 10 L/min and the core-sampled flow into the
APi was 0.8 L/min, with the remaining flow being discarded.



We calibrated the APi-TOF using trioctylmethylammonium bis (trifluoromethylsulfonyl)
imide (TBMA, $C_{27}H_{54}F_6N_2O_4S_2$) to facilitate the exact ion mass determination in both
positive and negative ion modes. We employed two calibration methods, the first one by
nebulizing TBMA and separating cluster ions with a high-resolution ultra-fine differential
mobility analyzer (UDMA) (see Steiner et al. (2014) for more information); the second one
by using electrospray ionization of a TBMA solution. The calibration with the electrospray
ionization was performed three times, one for each temperature. These calibrations enabled
mass/charge ($m/z$) measurements with high accuracy up to 1500 Th in the positive ion mode
and 900 Th in the negative ion mode.
Additionally, two peaks in the positive ion mode were identified as contaminants and also
used for calibration purposes at the three different temperatures: $C_{10}H_{14}OH^+$ and $C_{20}H_{28}O_2H^+$.
These peaks were present before the addition of ozone in the chamber (therefore being most
likely not products of α-pinene ozonolysis) and were also detected by a proton transfer
reaction time of flight mass spectrometer (PTR-TOF-MS). Both peaks appeared at the same
$m/z$ at all three temperatures. Therefore, based on the calibrations with the UDMA, the
electrospray and the two organic calibration peaks, we expect an accurate mass calibration at
the three temperatures.
2.3. **Experimental conditions**
All ambient ion composition data reported here were obtained during nucleation
experiments from pure α-pinene ozonolysis. The experiments were conducted under dark
conditions, at a relative humidity (RH) of 38% with an $O_3$ mixing ratio between 33 and 43
ppbv (Table 1). The APi-TOF measurements were made under both galactic cosmic ray
(GCR) and $\pi^+$ beam conditions, with ion-pair concentrations around 700 cm$^{-3}$ and 4000 cm$^{-3}$,
respectively.







Table 1. Experiments performed at the CLOUD chamber.

| Campaign | Experiment No. | Ionization | α-pinene (pptv) | $O_3$ (ppbv) | Mass spectrometer polarity | Temperature (˚C) |
|---|---|---|---|---|---|---|
| CLOUD 8 | 1211.02 | GCR | 258 | 33.8 | Negative | 5 |
| CLOUD 10 | 1710.04 | $\pi^+$ beam | 618 | 41.5 | Positive | 5 |
| CLOUD 10 | 1712.04 | $\pi^+$ beam | 511 | 40.3 | Negative and positive | 25 |
| CLOUD 10 | 1727.04 | $\pi^+$ beam | 312 | 43.3 | Negative and positive | -25 |


### 2.4. Quantum chemical calculations

Quantum chemical calculations were performed on the cluster ion formation from the
oxidation products of α-pinene. The Gibbs free energies of formation of representative HOM
clusters were calculated using the MO62X functional (Zhao and Truhlar, 2008), and the 6-
31+G(d) basis set (Ditchfield, 1971) using the Gaussian09 program (Frisch et al., 2009). This
method has been previously applied for clusters containing large organic molecules (Kirkby
et al., 2016).

## 3.  Results and discussion

### 3.1. Ion composition

Under relatively dry conditions, the main detected positive ions were $N_2H^+$ and $O_2^+$. With
increasing  RH we observed the water clusters $H_3O^+$, $(H_2O)\cdot H_3O^+$ and $(H_2O)_2\cdot H_3O^+$ as well as
$NH_4^+$, $C_5H_5NH^+$ (protonated pyridine), $Na^+$, and $K^+$. The concentrations of the precursors of
some of the latter ions are expected to be very low: for example, $NH_3$ mixing ratios were
previously found to be in the range of 0.3 pptv (at −25 ˚C), 2 pptv (at 5 ˚C)  and 4.3 pptv (at
25 ˚C) (Kürten et al., 2016). For the negative ions, $NO_3^-$ was the main detected background
signal. Before adding any trace gas to the chamber the signal of $HSO_4^-$ was at a level of 1%
of the $NO_3^-$ signal (corresponding to $<5\cdot10^{-4}$ molecules $cm^{-3}$, Kirkby et al., 2016), excluding
any contribution of sulfuric acid to nucleation in our experiments.




After initiating α-pinene ozonolysis, more than 460 organic ions were identified in the
positive spectrum. The majority of peaks were clustered with $NH_4^+$, while only 10.2 % of the
identified peaks were composed of protonated organic molecules. In both cases the organic
core was of the type $C_{7-10}H_{10-16}O_{1-10}$ for the monomer region and $C_{17-20}H_{24-32}O_{5-19}$ for the
dimer region.
In the negative spectrum we identified more than 530 HOMs, of which ~62%
corresponded to organic clusters with $NO_3^-$ or, to a lesser degree, $HNO_3 \cdot NO_3^-$. The rest of
the peaks were negatively charged organic molecules. In general, the organic core of the
molecules was of the type $C_{7-10}H_{9-16}O_{3-12}$ in the monomer region and $C_{17-20}H_{19-32}O_{10-20}$ in the
dimer region. For brevity we refer to the monomer, dimer (and n-mer) as $C_{10}$, $C_{20}$ and $C_{(10n)}$
respectively. Here, the subscript indicates the maximum number of carbon atoms in these
molecules, even though the bands include species with slightly fewer carbon atoms.

### 3.1.1.      Positive spectrum

The positive spectrum is characterized by bands of high intensity at $C_{20}$ intervals, as
shown in Figure 1B. Although we detected the monomer band ($C_{10}$), its integrated intensity
was much lower than the $C_{20}$ band; furthermore, the trimer and pentamer bands were almost
completely absent. Based on chemical ionization mass spectrometry measurements, Kirkby et
al. (2016) calculated that the HOM molar yield at 5°C was 3.2% for the ozonolysis of α-
pinene, with a fractional yield of 10 to 20% for dimers. A combination reaction of two
oxidized peroxy radicals has been previously reported to explain the rapid formation of
dimers resulting in covalently bound molecules (see Section 3.3). The pronounced dimer
signal with $NH_4^+$ indicates that (low-volatility) dimers are necessary for positive ion
nucleation and initial growth.  We observe growth by dimer steps up to $C_{80}$ and possibly even
$C_{100}$. A cluster of two dimers, $C_{40}$, with a mass/charge in the range of ~ 700 - 1100 Th, has a
mobility diameter around 1.5 nm (based on Ehn et al. (2011)).
Our observation of HOMs-$NH_4^+$ clusters implies strong hydrogen bonding between
the two species. This is confirmed by quantum chemical calculations which shall be
discussed in Section 3.3. Although hydrogen bonding could also be expected between HOMs
and $H_3O^+$, we do not observe such clusters.  This probably arises  from the higher proton





affinity of $NH_3$, 203.6 kcal/mol, compared with $H_2O$, 164.8 kcal/mol (Hunter and Lias,

1998). Thus, most $H_3O^+$ ions in CLOUD will transfer their proton to $NH_3$ to form $NH_4^+$.

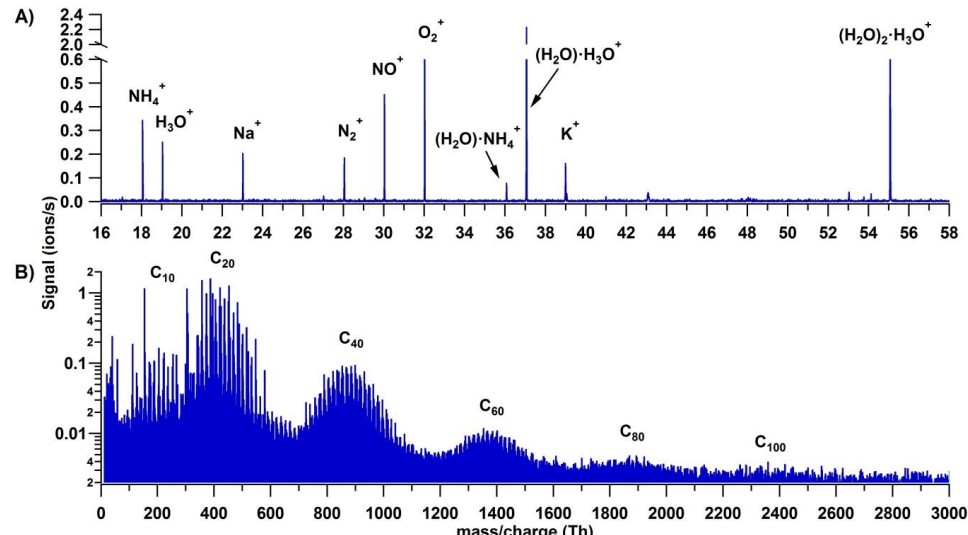

Figure 1. Positive spectra at 5˚C. A) Low mass region, where primary ions from galactic cosmic ray are observed, as well as secondary ions such as $NH_4^+$ which are formed by charge transfer to contaminants. B) Higher mass region during pure biogenic nucleation, which shows broad bands in steps of $C_{20}$. Most of the peaks represent clusters with $NH_4^+$.

### 3.1.2.  Negative spectrum

In the negative spectra, the monomer, dimer and trimer bands are observed during

nucleation (Fig. 2). Monomers and dimers have similar signal intensities, whereas the trimer

intensity is at least 10 times lower (Figure 2A and B). The trimer signal is reduced since it is

a cluster of two gas phase species ($C_{10}+C_{20}$). Additionally, a lower transmission in the APi-

TOF may also be a reason for the reduced signal.

In Fig. 2, we compare the CLOUD negative-ion spectrum with the one from nocturnal

atmospheric measurements from the boreal forest at Hyytiälä as reported by Ehn et al. (2010).

Panels 2A and 2B show the negative spectrum of α-pinene ozonolysis in the CLOUD

chamber on logarithmic and linear scales, respectively. Panel 2C shows the Hyytiälä

spectrum for comparison. Although the figure shows unit mass resolution, the high resolution





analysis confirms the identical composition for the main peaks: $C_8H_{12}O_7 \cdot NO_3^-$,
$C_{10}H_{14}O_7 \cdot NO_3^-$, $C_{10}H_{14}O_8 \cdot NO_3^-$, $C_{10}H_{14}O_9 \cdot NO_3^-$, $C_{10}H_{16}O_{10} \cdot NO_3^-$ and $C_{10}H_{14}O_{11} \cdot NO_3^-$
(marked in the monomer region), and $C_{19}H_{28}O_{11} \cdot NO_3$, $C_{20}H_{32}O_{13} \cdot NO_3^-$, $C_{19}H_{28}O_{13} \cdot NO_3^-$,
$C_{20}H_{32}O_{13} \cdot NO_3^-$, $C_{20}H_{30}O_{14} \cdot NO_3^-$, $C_{20}H_{30}O_{14} \cdot NO_3^-$, $C_{20}H_{30}O_{16} \cdot NO_3^-$ and $C_{20}H_{30}O_{18} \cdot NO_3^-$
(marked in the dimer region). The close correspondence in terms of composition of the main
HOMs from the lab and the field both in the monomer and dimer region indicates a close
reproduction of the atmospheric night-time conditions at Hyytiälä by the CLOUD
experiment. In both cases the ion composition was dominated by HOMs clustered with $NO_3^-$.
However, Ehn et al. (2010) did not report nocturnal nucleation, possibly because of a higher
ambient condensation sink than in the CLOUD chamber.

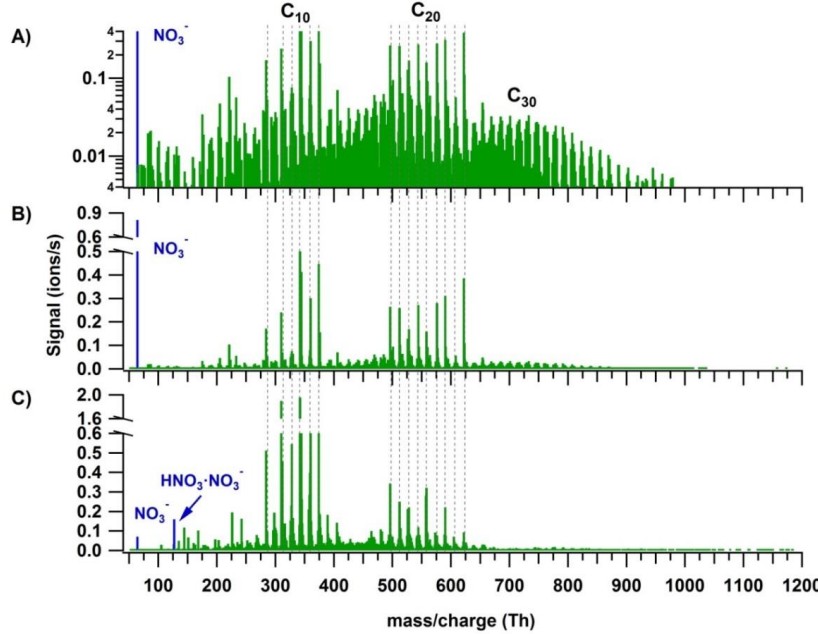


Figure 2. Comparison of the negative ion composition during α-pinene ozonolysis in CLOUD
and during night time in the boreal forest at Hyytiälä (Finland).  A) CLOUD spectrum on a
logarithmic scale. B) CLOUD spectrum on a linear scale. C) Typical night time spectrum
from the boreal forest at Hyytiälä (Finland), adapted from Ehn et al. (2010).



### 3.2. Temperature dependence


Experiments at three different temperatures (25 °C, 5°C and -25 ˚C) were conducted
at similar relative humidity and ozone mixing ratios (Table 1 and Figure 3). Mass defect plots
are shown for the same data in Figure 4. The mass defect is the difference between the exact
and the integer mass and is shown on the y-axis versus the mass/charge on the x-axis. Each
point represents a distinct atomic composition of a molecule or cluster. Although the
observations described in the following are valid for both polarities, the positive ion mode
shows the differences in the chemistry at the three temperatures more clearly.
The first point to note is the change in the distribution of the signal intensity seen in
Figure 3 (height of the peaks) and in Figure 4 (size of the dots) with temperature. In the
positive ion mode, the dimer band has the highest intensity at 25 and 5˚C, while at -25˚C the
intensity of the monomer becomes comparable to that of the dimer. This indicates a reduced
rate of dimer formation at -25 ˚C, or that the intensity of the ion signal depends on both the
concentration of the neutral compound and on the stability of the ion cluster. Although the
monomer concentration is higher than that of the dimers (Tröstl et al., 2016), the $C_{20}$ ions are
the more stable ion clusters as they can form more easily two hydrogen bonds with $NH_4^+$ (see
Section 3.3). Thus, positive clusters formed from monomers may not be stable enough at
higher temperatures. Moreover, charge transfer to dimers is also favored.



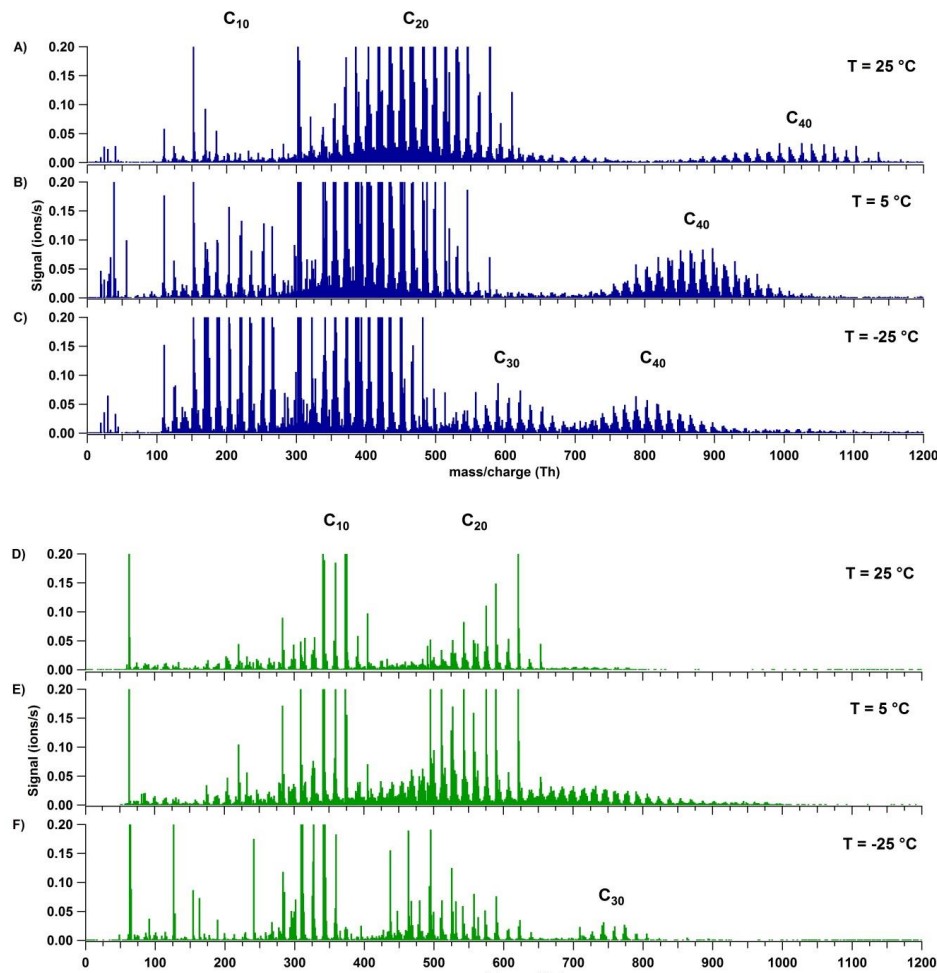

Figure 3. Positive (A-C) and negative (D-F) mass spectra during pure biogenic nucleation
induced by ozonolysis of □-pinene) at three temperatures: 25˚C (A, D), 5˚C (B, E) and -25˚C
(C, F). A progressive shift towards a lower oxygen content and lower masses is observed in
all bands as the temperature decreases. Moreover, the appearance of $C_{30}$ species can be seen
in the positive spectrum at the lowest temperature (C).

The data also show a "shift" in all band distributions towards higher masses with

increasing temperature, denoting a higher concentration of the more highly oxygenated
molecules and the appearance of progressively more oxygenated compounds at higher
temperatures. The shift is even more pronounced in the higher mass bands, as clearly seen in



the $C_{40}$ band of the positive ion mode in Figure 3 (A-C). In this case the combination of two
HOM dimers to a $C_{40}$ cluster essentially doubles the shift of the band towards higher
mass/charge at higher temperatures compared to the $C_{20}$ band. Moreover, the width of each
band increases with temperature, as clearly seen in the positive ion mode in Figure 4,
especially for the $C_{40}$ band. At high temperatures, the production of more highly oxygenated
HOMs seems to increase the possible combinations of clusters, resulting in a wider band
distribution.

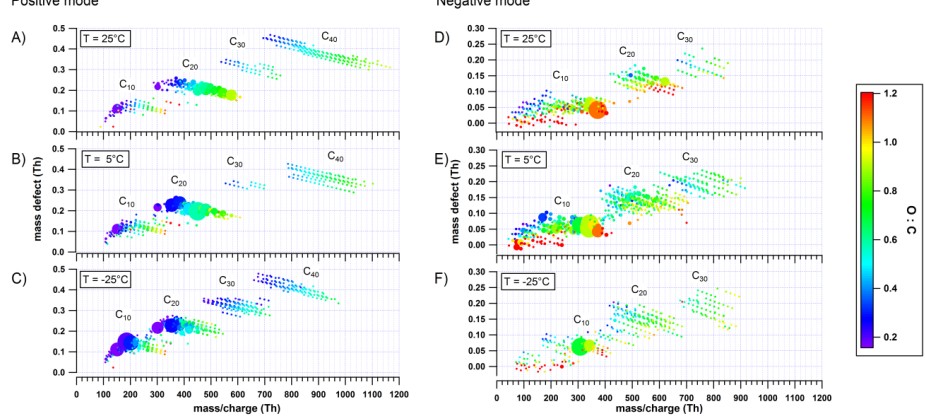


Figure 4. Mass defect plots with the color code denoting the O:C ratio (of the organic core) at
25, 5 and -25°C for positive (A-C) and negative ion mode (D-F). A lower O:C ratio is
observed in the positive ion mode than in the negative ion mode. The intensity of the main
peaks (linearly proportional to the size of the dots) changes with temperature for both
polarities due to a lower degree of oxygenation at lower temperature.

This trend in the spectra indicates that the unimolecular autoxidation reaction

accelerates at higher temperatures in competition to the bimolecular termination reactions
with $HO_2$ and $RO_2$. This is expected. If unimolecular and bimolecular reactions are
competitive, the unimolecular process will have a much higher barrier because the pre-
exponential term for a unimolecular process is a vibrational frequency while the pre-
exponential term for the bimolecular process is at most the bimolecular collision frequency,
which is four orders of magnitude lower. Quantum chemical calculations determine




activation energies between 22.56 and 29.46 kcal/mol for the autoxidation of different $RO_2$
radicals from α-pinene  (Rissanen et al., 2015). Thus, such a high barrier will strongly reduce
the autoxidation rate at the low temperatures.

The change in the rate of autoxidation is also reflected in the O:C ratio, both in the

positive ion mode (Figure 4 A-C), and the negative ion mode (D-F), showing a clear increase
with increasing temperature. The average O:C ratios (weighted by the peak intensities) are
presented in Table 2 for both polarities and the three temperatures, for all the identified peaks
(total) and separately for the monomer and dimer bands For a temperature change from 25 to
-25°C the O:C ratio decreases for monomers, dimers and total number of peaks. At high
masses (e.g., for the $C_{30}$ and $C_{40}$ bands), the O:C ratio may be slightly biased since accurate
identification of the molecules is less straightforward: as an example, $C_{39}H_{56}O_{25} \cdot NH_4^+$ has an
exact mass of 942.34 Th (O/C = 0.64), which is very similar to $C_{40}H_{60}O_{24} \cdot NH_4^+$ at 942.38 Th
(O/C = 0.60). However, such possible misidentification would not influence the calculated
total O/C by more than 0.05, and the main conclusions presented here remain robust.

Table 2. Signal weighted average O:C ratios for positive and negative
spectra at 25, 5 and -25 °C.

| Temperature (°C) | O/C | | | | | |
| --- | --- | --- | --- | --- | --- | --- |
| | Positive mode | | | Negative mode | | |
| | Monomer | Dimer | Total | Monomer | Dimer | Total |
| 25 | 0.37 | 0.57 | 0.54 | 0.94 | 0.81 | 0.90 |
| 5 | 0.34 | 0.51 | 0.49 | 0.88 | 0.66 | 0.75 |
| -25 | 0.31 | 0.38 | 0.36 | 0.79 | 0.65 | 0.68 |


The O:C ratios are higher for the negative ions than for the positive ions at any of the

three temperatures. Although some of the organic cores are the same in the positive and
negative ion mode, the intensity of the peaks of the most oxygenated species is higher in the
negative spectra. While the measured O:C ratio ranges between 0.4 and 1.2 in the negative
ion mode, it is between 0.1 and 1.2 in the positive ion mode. An O:C ratio of 0.1, which was
detected only in the positive ion mode, corresponds to monomers with 1 oxygen atom or
dimers with two oxygen atoms. The presence of molecules with such low oxygen content was





also confirmed with a proton transfer reaction time-of-flight mass spectrometer (PTR-TOF-
MS), at least in the monomer region. The ions with this low O:C ratio are probably from the
known main oxidation products like pinonaldehyde, pinonic acid, etc. It is likely that these
molecules, which were detected only in the positive mode, contribute only to the growth of
the newly formed particles (if at all) rather than to nucleation, owing to their high volatility
(Tröstl et al., 2016). In this sense, the positive spectrum could reveal both the molecules that
participate in the new particle formation and those that contribute to growth. The differences
in the O:C ratios between the two polarities are a result of the affinities of the organic
molecules to form clusters either with $NO_3^-$ or $NH_4^+$, which, in turn, depends on the
molecular structure and the functional groups. Hyttinen et al. (2015) reported the binding
energies of selected highly oxygenated products of cyclohexene detected by a nitrate CIMS,
finding that the addition of OOH groups to the HOM strengthens the binding of the organic
core with $NO_3^-$. Even when the number of H-bonds between $NO_3^-$ and HOM remains the
same, the addition of more oxygen atoms to the organic compound could strengthen the
bonding with the $NO_3^-$ ion. Thus, the less oxygenated HOMs were not detected in those
experiments, neither in ours, in the negative mode. The binding energies were calculated for
the positive mode HOMs-$NH_4^+$ and are discussed in Section 3.3.

We also tested to which extent the formation of the $C_{40}$ band could be reproduced by

permutation of the potential $C_{20}$ clusters weighted by the dimer signal intensity. Figure 5
shows the measured spectrum (blue) and two types of modeled tetramers: one combining all
peaks from the $C_{20}$ band (light gray) and one combining only those peaks with an organic
core with O/C $\geq$ 0.4, i.e. likely non-volatile molecules (dark gray). The better consistency of
the latter with the measured tetramer band suggests that only the molecules with O/C $\geq$ 0.4
are able to form the tetramer cluster. This would mean that $C_{20}$ molecules with 2-7 oxygen
atoms are likely not to contribute to the nucleation, but only to the growth of the newly
formed particles.





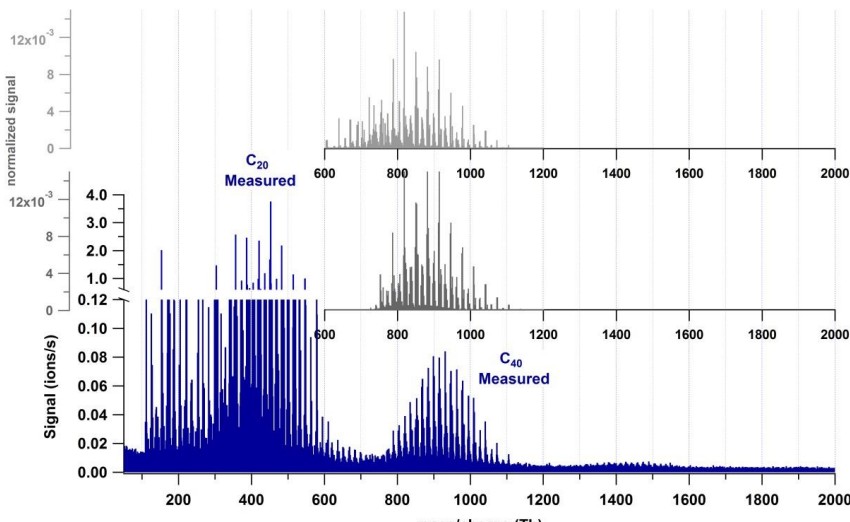


Figure 5. Comparison of the positive ion mode spectrum measured (blue), the $C_{40}$ band obtained by the combination of all $C_{20}$ molecules (light gray) and the $C_{40}$ band obtained by combination of only the $C_{20}$ molecules with $O/C \geq 0.4$ (dark gray). The low or absent signals at the lower masses obtained by permutation suggests that only the highly oxygenated dimers are able to cluster and form $C_{40}$.

These two observations (change in signal distribution and band "shift") are not only valid for positive and negative ions, but also for the neutral molecules as observed by two nitrate chemical-ionization atmospheric-pressure-interface time-of-flight mass spectrometers (CI-APi-TOF; Aerodyne Research Inc. and Tofwerk AG). This confirms that there is indeed a change in the HOM composition with different temperature rather than a charge redistribution effect which would only be observed for the ions (APi-TOF). The detailed analysis of the neutral molecules detected by these CI-APi-TOFs will be subject of another paper and is not discussed here.

A third distinctive trend in the positive mode spectra at the three temperatures is the increase in signal intensity of the $C_{30}$ band at -25˚C. The increase in the signal of the trimer also seems to occur in the negative ion mode when comparing panels D and F in Figure **3**. For this polarity, data from two campaigns were combined (Table **1**). To avoid a bias by possible differences in the APi-TOF settings, we only compare the temperatures from the same campaign, CLOUD 10, therefore experiments at 25 ˚C and -25 ˚C. The increase in the





trimer signal may be due to greater stability of the monomer-dimer clusters or even of three
$C_{10}$ molecules at low temperatures, as further discussed below.

3.3. **Quantum chemical calculations**
Three points were addressed in the quantum chemical calculations to elucidate the most
likely formation pathway for the first clusters, and its temperature dependence. These
included (i) the stability of the organic cores with $NO_3^-$ and $NH_4^+$ depending on the binding
functional group, (ii) the difference between charged and neutral clusters in terms of
clustering energies, and finally (iii) the possible nature of clusters in the dimer and trimer
region.
The calculations showed that among the different functional groups the best interacting
groups with $NO_3^-$ are in order of importance carboxylic acids (R–C(=O)–OH), hydroxyls (R–
OH), peroxy acids (R–C(=O)–O–OH), hydroperoxides (R–O–OH) and carbonyls ( R–(R′–)
C=O). On the other hand, $NH_4^+$ preferably forms a hydrogen bond with the carbonyl group
independent of which functional group the carbonyl group is linked to: Figure 6 shows
examples of $NH_4^+$ clusters with corresponding free energies of formation for carbonyls ($\Delta G$=
-17.98 kcal/mol), carboxylic acid ($\Delta G$= -17.32 kcal/mol), and peroxy acid ($\Delta G$= -17.46
kcal/mol). For the three examples shown, the interaction of one hydrogen from $NH_4^+$ with a
C=O group is already very stable with a free energy of cluster ion formation close to -18
kcal/mol.





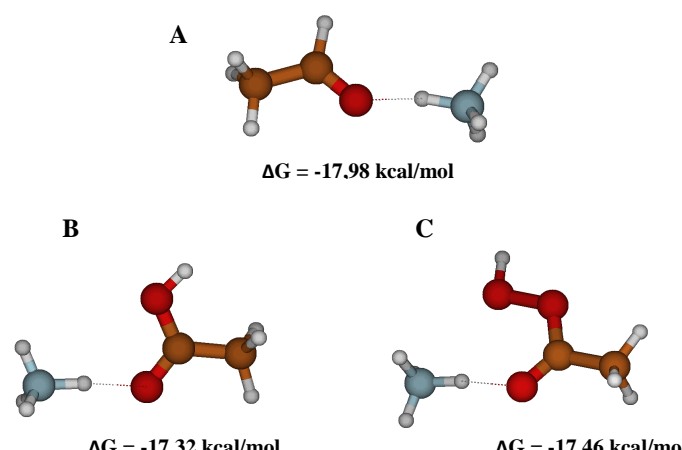

Figure 6. Quantum chemical calculations of the free energy related to the cluster formation between $NH_4^+$ and three structurally similar molecules with different functional groups: A) acetaldehyde, B) acetic acid and C) peracetic acid.




To evaluate the effect of the presence of a second C=O to the binding of the organic
compound with $NH_4^+$, we performed a series of calculations with a set of surrogates
containing two C=O groups separated by a different number of atoms, as shown in Figure 7.
The addition of a second functional group allows the formation of an additional hydrogen-
bond, increasing the stability of the cluster considerably (almost two folds) from about -18
kcal/mol to -34.07 kcal/mol, whereby the position of the second functional group to form an
optimal hydrogen bond (with a 180° angle for N-H-O) strongly influences the stability of the
cluster, as can be seen in Figure 7. Thus, optimal separation and conformational flexibility of
functional groups is needed to enable an effective formation of two hydrogen bonds with
$NH_4^+$. This could be an explanation for the observation that the signal intensity is higher for
dimers than for monomers, as dimers can more easily form two optimal hydrogen bonds with
$NH_4^+$.






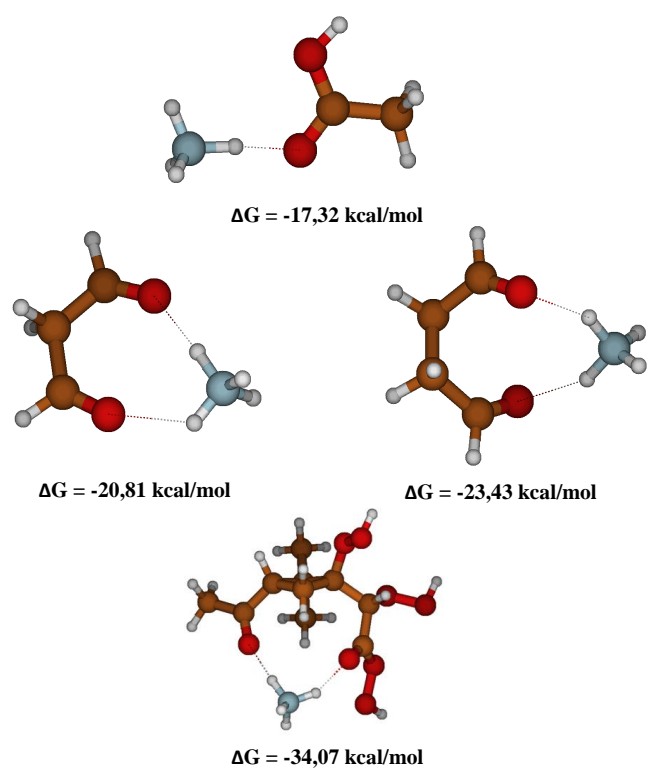

**ΔG = -17,32 kcal/mol**

**ΔG = -20,81 kcal/mol**          **ΔG = -23,43 kcal/mol**

**ΔG = -34,07 kcal/mol**


Figure 7. Quantum chemical calculations for different organic molecules with a carbonyl as the interacting functional group with $NH_4^+$. Increasing the interacting groups from one to two increases the stability of the cluster. The distance between the interacting groups also influences the cluster stability.



As shown by Kirkby et al. (2016), ions increase the nucleation rates by one to two
orders of magnitudes compared to neutral nucleation. This is expected due to the strong
electrostatic interaction between charged clusters. To understand how the stability difference
relates to the increase in the nucleation rate, the ΔGs of charged and neutral clusters were
compared. For this, $C_{10}H_{14}O_7$ and $C_{20}H_{30}O_{14}$ were selected as representative molecules of the
monomer and dimer region, respectively (Kirkby et al., 2016). Table 3 shows the calculated
free energies of formation (ΔG) of neutral, positive and negative clusters from these $C_{10}$ and
$C_{20}$ molecules at the three temperatures of the experiment. Results show that at 5°C, for



example, $\Delta G$ of the neutral dimer ($C_{10} + C_{10}$) is -5.76 kcal/mol while it decreases to -20.95
kcal/mol when a neutral and a negative ion form a cluster ($C_{10} + C_{10}^-$). Similarly, trimers
show a substantial increase in stability when they are charged, i.e., from -2.15 kcal/mol to
-19.9 kcal/mol, for the neutral and negative cases, respectively. The reduced values of $\Delta G$ for
the charged clusters (positive and negative) indicate a substantial decrease in the evaporation
rate compared to that for neutral clusters, and, therefore, higher stability. Comparing the
$NH_4^+$ and $NO_3^-$ clusters, the energies of formation for the monomer are -22.5 kcal/mol and
-25.99 kcal/mol, respectively, showing slightly higher stability for the negative cluster.
Inversely, the covalently bound dimer showed greater stability for the positive ion (-30.9
kcal/mol) compared to the negative ion (-25.65 kcal/mol).

Table 3. Gibbs free energies of cluster formation $\Delta G$ at three different temperatures. $\Delta G$ for
the molecules $C_{10}H_{14}O_7$ ($C_{10}$) and $C_{20}H_{30}O_{14}$ ($C_{20}$) forming neutral, as well as negative and
positive ion clusters.

|  | Cluster process | $\Delta G_{-25°C}$ (kcal/mol) | $\Delta G_{5°C}$ (kcal/mol) | $\Delta G_{25°C}$ (kcal/mol) |
|---|---|---|---|---|
| Neutral | $C_{10} + C_{10}$ | -7.33 | -5.76 | -4.70 |
|  | $C_{10} + C_{20}$ | -3.28 | -2.15 | -1.39 |
| Positive | $C_{10} + NH_4^+$ | -23.40 | -22.50 | -21.80 |
|  | $C_{20} + NH_4^+$ | -31.80 | -30.90 | -30.20 |
|  | $C_{10} + C_{10} \cdot NH_4^+$ | -12.90 | -11.70 | -10.90 |
|  | $C_{20} + C_{10} \cdot NH_4^+$ | -26.00 | -24.30 | -23.30 |
|  | $C_{10} + C_{20} \cdot NH_4^+$ | -17.60 | -15.90 | -14.80 |
| Negative | $C_{10} + C_{10}^-$ | -22.22 | -20.95 | -20.09 |
|  | $C_{20} + C_{10}^-$ | -21.36 | -19.90 | -18.91 |
|  | $C_{10} + NO_3^-$ | -27.27 | -25.99 | -25.14 |
|  | $C_{20} + NO_3^-$ | -26.97 | -25.65 | -24.75 |
|  | $C_{10} + C_{10} \cdot NO_3^-$ | -11.34 | -10.09 | -9.25 |


The temperature dependence of cluster formation is shown in Figure **8** for the positive ion
clusters. The blue and brown solid lines represent the needed $\Delta G$ for evaporation-collision
equilibrium at 0.3 pptv and 1 pptv HOM mixing ratio, respectively, calculated as described
by Ortega et al. (2012). The markers show the calculated formation enthalpies $\Delta G$ for each
of the possible clusters. For all cases, the trend shows an evident decrease in $\Delta G$ with
decreasing temperature, with a correspondingly reduced evaporation rate.



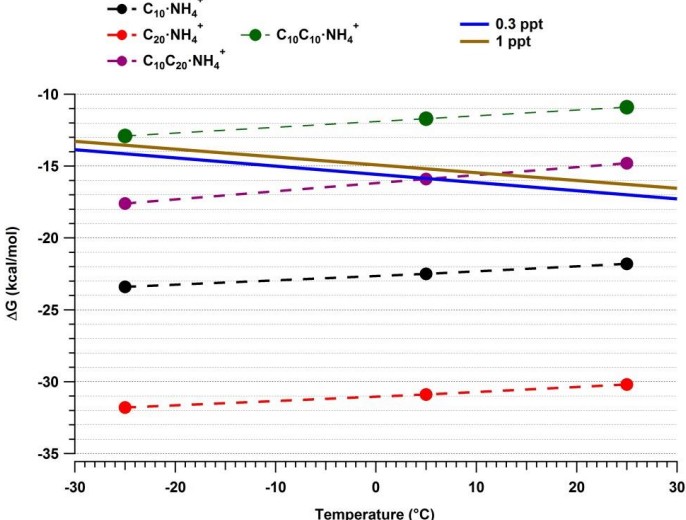


Figure 8. Quantum chemical calculations of Gibbs free energies for cluster formation at -25, 5 and 25˚C. Solid lines represent the required $\Delta G$ for equilibrium between evaporation and collision rates at 0.3 pptv and 1 pptv of the HOM mixing ratio, respectively. Markers show the $\Delta G$ for each cluster (organic core clustered with $NH_4^+$) at the three temperatures. $C_{10} \cdot NH_4^+$ (black circles) represent the monomer, $C_{20} \cdot NH_4^+$ (red circles) represent the covalently bound dimer, $C_{10}C_{10} \cdot NH_4^+$ (green circles) represent the dimer formed by the clustering of two monomers and $C_{10}C_{20} \cdot NH_4^+$ (purple circles) denote the preferential pathway for the trimer cluster (see Table 3).

At all three temperatures, the monomer cluster $C_{10} \cdot NH_4^+$ falls well below the equilibrium

lines, indicating high stability. Even though the difference between -25˚C and 25˚C is just

-1.6 kcal/mol in free energy, it is enough to produce a substantial difference in the intensity of

the band, increasing the signal at least 8-fold at -25˚C (as discussed in Section 3.2). In the

case of the dimers, we consider the possibility of their formation by collision of a monomer

$C_{10} \cdot NH_4^+$ with another $C_{10}$ (resulting in a $C_{10}C_{10} \cdot NH_4^+$ cluster) or the dimer as $C_{20} \cdot NH_4^+$

cluster. The calculations show clearly that the cluster $C_{10}C_{10} \cdot NH_4^+$ is not stable at any of the

three temperatures (green line). In contrast, the covalently-bound $C_{20}$ forms very stable

positive and negative ion clusters (see Table 3). Trimers are mainly observed at lower

temperatures. Since the $C_{10}C_{10} \cdot NH_4^+$ cluster is not very stable, we discard the possibility of a

trimer formation of the type $C_{10}C_{10}C_{10} \cdot NH_4^+$. Thus, the trimer is likely the combination of a

monomer and a covalently-bound dimer ($C_{20}C_{10} \cdot NH_4^+$). According to our calculations (Table



3) the preferred evaporation path for this cluster is the loss of $C_{10}$ rather than the evaporation
of $C_{20}$. Therefore, we have chosen to represent only this path in Figure **8**. The □G of this
cluster crosses the evaporation-condensation equilibrium around 5 ˚C and 14 ˚C for a HOM
mixing ratio of 0.3 pptv and 1 pptv, respectively, in good agreement with the observed signal
increase of the trimer at -25°C (Figure 3 A-C). It is important to note that, due to the
uncertainty in the calculations, estimated to be ≤ 2 kcal/mol, we do not consider the crossing
as an exact reference.

The $\Delta G$ of the negative ion clusters, which are also presented in Table 3, decrease

similarly to the positive ion clusters by around 2 kcal/mol between 25˚C and -25˚C. The
cluster formation energies of the monomer and the dimer with $NO_3^-$ are in agreement with the
observed comparable signal intensity in the spectrum (Figure 2) in a similar way as the
positive ion clusters. The covalently-bonded dimer ion $C_{20} \cdot NO_3^-$ is also more stable
compared to the dimer cluster $C_{10}C_{10} \cdot NO_3^-$, suggesting that the observed composition results
from covalently bonded dimers clustering with $NO_3^-$ rather than two individual $C_{10}$ clustering
to form a dimer.

The formation of a covalently bonded trimer seems unlikely, so the formation of highly

oxygenated molecules is restricted to the monomer and dimer region. The trimer could result
from the clustering of $C_{10}$ and $C_{20}$ species. Similarly, and based on the $C_{20}$ pattern observed in
Figure 1B, we believe that the formation of the tetramer corresponds to the collision of two
dimers. No calculations were done for this case due to the complexity related to the sizes of
the molecules, which prevents feasible high level quantum chemical calculations.

Finally, the comparison between the $\Delta G$ values for charged and neutral clusters as

presented in Table 3 confirms the expected higher stability of ion clusters, decreasing the
evaporation rate of the nucleating clusters and enhancing new particle formation.

## 4. Conclusions

Ions observed during pure biogenic ion-induced nucleation were comprised of mainly

organics clustered with $NO_3^-$ and $NH_4^+$ and to a lesser extent charged organic molecules only
or organics clustered with $HNO_3NO_3^-$. We found good correspondence between the negative



ions measured in CLOUD with those observed in the boreal forest of Hyytiälä. The observed
similarity in the composition of the HOMs in the monomer and dimer region during new-
particle formation experiments at CLOUD suggests that pure biogenic nucleation might be
possible during night time if the condensation sink is sufficiently low, i.e., comparable to that
in the CLOUD chamber, where the wall loss rate for $H_2SO_4$ is $1.8 \cdot 10^{-3}$ s$^{-1}$ (Kirkby et al.,
2016). The positive mass spectrum showed a distinctive pattern corresponding to progressive
addition of dimers ($C_{20}$), up to cluster sizes in the range of stable small particles.
Temperature strongly influenced the composition of the detected molecules in several
ways. With increasing temperature, a higher oxygen content (O:C ratio) in the molecules was
observed in both the positive and the negative mode. This indicates an increase in the
autoxidation rate of peroxy radicals which is in competition with their bimolecular
termination reactions with $HO_2$ and $RO_2$.
A broader range of organic molecules was found to form clusters with $NH_4^+$ than with
$NO_3^-$. Quantum chemical calculations using simplified molecules show that $NH_4^+$ preferably
forms a hydrogen bond with a carbonyl group independently of other functional groups
nearby. The addition of a second hydrogen bond was found to increase the cluster stability
substantially. Thus, the $C_{20}$-ions are the more stable ion clusters as they can form more easily
two hydrogen bonds with $NH_4^+$. Although molecules with low oxygen content were
measured in the $C_{20}$ band (1 - 4 oxygen atoms), only the molecules with O/C $\geq$ 0.4 seem to be
able to combine to form larger clusters.
The quantum chemical calculations showed that the covalently-bonded dimer $C_{20} \cdot$
$NO_3^-$ is also more stable than the dimer cluster $C_{10}C_{10} \cdot NO_3^-$, suggesting that the observed
composition results from covalently bonded molecules clustering with $NO_3^-$ rather than $C_{10}$
clusters.
Temperature affected cluster formation by decreasing evaporation rates at lower
temperatures, despite of the lower O:C ratio. In the positive mode a pronounced growth of
clusters by addition of $C_{20}$-HOMs was observed. The formation of a $C_{30}$-cluster only
appeared at the lowest temperature, which was supported by quantum chemical calculations.
In the negative mode it appeared as well that the signal of the $C_{30}$-clusters became stronger
with lower temperature. The $C_{40}$- and higher clusters were probably not seen because of too





low sensitivity in this mass range due to the applied instrumental settings. More
measurements are needed to determine if the cluster growth of positive and negative ions
proceeds in a similar or different way.
**5. Acknowledgement**
We would like to thank CERN for supporting CLOUD with important technical and financial
resources, and for providing a particle beam from the CERN Proton Synchrotron. We also
thank P. Carrie, L.-P. De Menezes, J. Dumollard, K. Ivanova, F. Josa, I. Krasin, R. Kristic, A.
Laassiri, O. S. Maksumov, B. Marichy, H. Martinati, S. V. Mizin, R. Sitals, A. Wasem and
M. Wilhelmsson for their important contributions to the experiment. This research has
received funding from the EC Seventh Framework Programme (Marie Curie Initial Training
Network "CLOUD-ITN" no. 215072, MC-ITN "CLOUD-TRAIN" no. 316662, the ERC-
Starting grant "MOCAPAF" no. 57360, the ERC-Consolidator grant "NANODYNAMITE"
no. 616075 and ERC-Advanced grant "ATMNUCLE" no. 227463), European Union's
Horizon 2020 research and innovation programme under the Marie Sklodowska-Curie grant
agreement no. 656994, the PEGASOS project funded by the European Commission under the
Framework Programme 7 (FP7-ENV-2010-265148), the German Federal Ministry of
Education and Research (project nos. 01LK0902A and 01LK1222A), the Swiss National
Science Foundation (project nos. 200020_152907, 206021_144947 and 20FI20_159851), the
Academy of Finland (Center of Excellence project no. 1118615), the Academy of Finland
(135054, 133872, 251427, 139656, 139995, 137749, 141217, 141451, 299574), the Finnish
Funding Agency for Technology and Innovation, the Väisälä Foundation, the Nessling
Foundation, the University of Innsbruck research grant for young scientists (Cluster
Calibration Unit), the Portuguese Foundation for Science and Technology (project no.
CERN/FP/116387/2010), the Swedish Research Council, Vetenskapsrådet (grant 2011-5120),
the Presidium of the Russian Academy of Sciences and Russian Foundation for Basic
Research (grants 08-02-91006-CERN and 12-02-91522-CERN), the U.S. National Science
Foundation (grants AGS1447056, and AGS1439551), and the Davidow Foundation. We
thank the *tofTools* team for providing tools for mass spectrometry analysis.





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
