# Peer review of "Influence of temperature on the molecular composition of ions and charged clusters during pure biogenic nucleation"

_Atmospheric Chemistry and Physics, 2017_

## Referee Comment (RC1) · Anonymous Referee #1 · 28 Jun 2017

General comments

The manuscript presents data from pure $\alpha$-pinene ozonolysis nucleation experiments performed at three different temperatures at the CERN CLOUD chamber. Organic ions were measured with an APi-ToF. The authors discuss differences in positive and negative ion mass spectra, as well as ion binding energies for the different temperature conditions. This is a very well written paper. It presents information on atmospheric particle nucleation at a range of tropospheric temperatures, and thus is of high atmospheric relevance. There are a few aspects that could profit from further clarification, and the manuscript leaves the reader somewhat in the dark of the implications of the

results. A paragraph discussing the importance of the observed differences in ion composition for atmospheric new particle formation is missing. I suggest publication after these aspects and the specific comments have been addressed.

Specific comments

P. 4, l. 120 – 122: It would be informative to give a rough idea of the actual tropospheric altitudes these temperatures correspond to, in regions where and seasons when $\alpha$-pinene emissions are important.

P. 5, l. 136 – 139: Does "before the start of the experiments" refer to each experiment, or a series of experiments/an entire campaign? Please clarify. If you mean a series of experiments - how do backgrounds evolve during their course?

P. 6, l. 158 – 174: Your description of calibrations refers to mass calibration only. Are there no sensitivity calibrations? Throughout the manuscript, instrument transmission is mentioned a few times, but not discussed specifically. See also comments further down – can you expect ion rates to be consistent across experiments with similar precursor concentrations? Can you expect the instrument to be able to measure potential closure between (simplified) e.g. a decrease in monomer rate due to an increase in dimer formation? A short paragraph on instrument limitations would help the reader put your ion rates into perspective.

P. 7, l. 197 - 205: Presumably this paragraph describes the chamber background before increasing RH or adding precursor gases. This could be stated more clearly. Was ionization already on? Please clarify. What does "relatively dry" mean. If possible replace by RH percentage.

P. 8, l. 207: In your spectra the majority of ions is in the form of clusters with NH4+ or NO3-, or in other words, contaminants. Does this mean your results depend on contamination of the chamber, and in a perfectly clean chamber you would miss a large fraction of your APi-ToF spectra? Please clarify.

P. 10, l. 262 – 264: Are differences in condensation sink the only possible explanation for the "missing" nocturnal new particle formation in Hyytiälä? How do precursor concentrations compare? Have you done similar experiments in the CLOUD chamber, but with condensation sink? Please elaborate shortly on the meaning of similarity in observed spectra/ion composition between field and laboratory, but difference in new particle formation rates.

P. 11, l. 276 – 278: Do you have an explanation for this observation? Could differences in NH3+ mixing ratios for the different temperatures play a role?

P. 11, l.279 – 280: This would be easier to see if y-axis ranges were expanded to beyond 0.2

P. 11, l. 279 – 288: Presumably these measurements are during nucleation/before the onset of particle growth, but can you rule out influence of condensation to the walls in this shift? And related, it is not well discernible from the figure that the rate of dimer formation at -25 ËŽC is reduced, there could also simply be an increase in monomer formation rate. Please clarify.

P. 15, l. 368 - p. 16, l. 379: What exactly is meant by "better consistency"? Overlapping of ion masses? Can one expect ion rate closure? Y-axes labels would suggest otherwise, but it is hard to see.

P. 22, l. 513: Shouldn't the sentence finish with "with lower temperature"?

P. 23, l. 544: You mention O:C ratio in a subclause. O:C ratios are being discussed as a determining factor in many processes related to atmospheric new particle number or mass formation. Your results imply temperature to be just as important. Please elaborate further on such an important implication.

Technical corrections

P. 8, l. 221 No reference to Figure 1A Figure 4 would profit from larger axis labels

P. 14, l. 329: Dot is missing

P. 22, l. 491: Delta is not printed properly
* * *

---

## Referee Comment (RC2) · Anonymous Referee #2 · 25 Jul 2017

The manuscript describes laboratory experiments in the CERN CLOUD chamber. The authors investigate the molecular composition of positive and negative HOM clusters measured with APi-TOFs at three different temperatures (25°C, 5°C and -25°C). The authors discuss the cluster formation in the positive and negative ion mode, the average oxygen-to-carbon (O:C) ratios focusing on the influence of the changing temperature. They essentially conclude a decrease in the rate of autoxidation with decreasing temperature. The experimental findings are supported by quantum chemical calculations of the binding energies of representative neutral and charged clusters. In general, the paper is well written and presents an interesting topic that is well suited to be published in ACP. The molecular processes of new particle formation, especially if organic

molecules are also involved, are not well understood. Therefore, I suggest to publish the manuscript in ACP after considering the following comments.

Page 6, line 167: Impurities from alpha-pinene or chamber background ? Could that be signals from previous experiments (e.g. pinanediol ?)

Page 7, line 197: What does "under relatively dry conditions" mean ?

Page 10, line 257,258: The ion C20H32O13.NO3- is mentioned two times ?

Page 14, line 329: ". . . bands For a . . ." !?

Page: 14, Table 2: The O/C-ratios from the monomers to the dimers are increasing in the positive mode and decreasing in the negative mode. While the decrease could be a result of oxygen loss in the formation of the dimers (in this case covalently bonded dimers (e.g. condensation reactions !?)), the increase in the O/C-ratio is more difficult to explain. One possibility would be the preferred formation from monomers with a rather high O/C-ratio, similar what the authors use as explanation for the tetramer formation, however, I wonder if the authors also included the C10H14OH+ signal in their signal weighted average O/C calculation (looking at figure 4 it seems so). In this case the inclusion of the background signal is of course misleading and should be corrected.

Page 14, line 341: The authors often use the expression "cores". I suggest to use simply "compounds" or "molecules".

Page 14, line 345: Again, I have the impression that the authors refer to the background signals mentioned earlier (compounds containing 1 oxygen atom) !? The explanation given on the next page (main oxidation products) is definitely not satisfying (in other words: pinonaldehyde is C10H16O2 and pinonic acid C10H16O3 – a factor of 2-3 higher in oxygen than needed).

---

## Author Comment (AC1) · 17 Oct 2017

*"Influence of temperature on the molecular composition of ions and charged clusters during pure biogenic nucleation" by Carla Frege et al.*

**Anonymous Referee #1**

General comments

The manuscript presents data from pure α-pinene ozonolysis nucleation experiments performed at three different temperatures at the CERN CLOUD chamber. Organic ions were measured with an APi-ToF. The authors discuss differences in positive and negative ion mass spectra, as well as ion binding energies for the different temperature conditions. This is a very well written paper. It presents information on atmospheric particle nucleation at a range of tropospheric temperatures, and thus is of high atmospheric relevance. There are a few aspects that could profit from further clarification, and the manuscript leaves the reader somewhat in the dark of the implications of the results. A paragraph discussing the importance of the observed differences in ion composition for atmospheric new particle formation is missing. I suggest publication after these aspects and the specific comments have been addressed.

We added a section on the implications at the end of the conclusion. See comments below.

Specific comments

P. 4, l. 120 – 122: It would be informative to give a rough idea of the actual tropospheric altitudes these temperatures correspond to, in regions where and seasons when α- pinene emissions are important.

We added the paragraph: *This spans the temperature range where NPF might occur in southern latitudes (25°C), high-latitude boreal regions (5°C) and free troposphere (-25°). For example, NPF events were reported to occur in an Australian Eucalypt forest (Suni, T. et al., Atmos. Chem. Phys., 8, 129-139, 10.5194/acp-8-129-2008, 2008) and at the boreal station in Hyytiälä (Kulmala, M et al, Science, 339, 943–946, 2013). New particle formation by organic vapors was also observed at a high mountain station (Bianchi et al. 2016). Moreover, high aerosol particle concentrations were measured in the upper troposphere over the Amazon Basin and attributed to the oxidation of biogenic volatile organic compounds.*

P. 5, l. 136 – 139: Does "before the start of the experiments" refer to each experiment, or a series of experiments/an entire campaign? Please clarify. If you mean a series of experiments - how do backgrounds evolve during their course?

"Before the start of the experiments" refers to the start of the campaign. This has been clarified.

 Your description of calibrations refers to mass calibration only. Are there no sensitivity calibrations? Throughout the manuscript, instrument transmission is mentioned a few times, but not discussed specifically. See also comments further down – can you expect ion rates to be consistent across experiments with similar precursor concentrations? Can you expect the instrument to be able to measure potential closure between (simplified) e.g. a decrease in monomer rate due to an increase in dimer formation? A short paragraph on instrument limitations would help the reader put your ion rates into perspective.

Yes, our calibration was a mass calibration only. We agree with the referee that a sensitivity calibration would have been interesting. However, this is difficult to achieve and needs specialized equipment. This needs to deliver a well-defined concentration of ions of different m/z. Since our measurements are inherently qualitative and not quantitative (i.e. concentrations) we did not perform a sensitivity calibration. We do expect ion rates to be consistent across experiments since the settings used during the whole campaign were the same; therefore the relative transmission of ions was kept constant. The transmission is the fraction of ions reaching the detector out of those entering the inlet. The transmission varies with m/z, see Junninen et al. (AMT, 3, 1039–1053, 2010) and Heinritzi et al. (AMT, 9, 1449-1460, 2016).

In theory you can reach a closure between a decrease of compounds and increase of others (Heinritzi et al., 2016). However, you would need to know the ionization and clustering energies of all the compounds to calculate their relative ion distribution. GCR or the $\pi^+$ beam generates a certain amount of ions in the chamber, which are then distributed among all the compounds, clusters and particles. Thus, due to lack of information on clustering energies it is not possible to determine the ion distribution.

We added a short paragraph on the characteristics of the measurements, which reads:

*There is no direct chemical ionization in front of the instrument. The APi-TOF measures the positive or negative ions and cluster ions as they are present in the ambient atmosphere. As described above in the CLOUD chamber ions are formed by GCR or deliberately by a π+ beam, leading to ion concentrations of a few hundred to thousands per $cm^3$, respectively. In our chamber the ionizing species are $NH_4^+$ and $NO_3^-$ (see below). These ions mainly form clusters with the organic molecules, which process is driven by the cluster energies. Therefore, the signals obtained do not provide a quantitative measure of the concentrations of the compounds. The higher the cluster energy with certain compounds the higher the ion cluster concentration will be.*

 Presumably this paragraph describes the chamber background before increasing RH or adding precursor gases. This could be stated more clearly. Was ionization already on? Please clarify. What does "relatively dry" mean. If possible replace by RH percentage.

This paragraph has been clarified. By "relative dry" we meant 0% RH. For the referee information we show below a plot with the increasing RH described in the paragraph and the appearance of $N_2H^+$ and $O_2^+$ and the water clusters. This was only with GCR, without the $\pi^+$ beam.

[Figure]

P. 8, l. 207: In your spectra the majority of ions is in the form of clusters with NH4+ or NO3-, or in other words, contaminants. Does this mean your results depend on contamination of the chamber, and in a perfectly clean chamber you would miss a large fraction of your APi-ToF spectra? Please clarify.

Indeed, the type and prevalence of the cluster ion depends on the type and degree of contamination. In a perfectly (100.0%) clean humid chamber the ionizing ions would be $H_3O^+$ and $O_2^-$. However, this is impossible to achieve. In a freshly cleaned chamber the levels of $NH_3$ and $HNO_3$ are expected to be way below 1 ppt. Despite these trace level concentrations, they become the main carriers of charge as also found in the real, relatively clean atmosphere (Frege et al. 2017; Ehn et al. 2010). In the ambient, most of the negative cluster ions are composed of an organic molecule clustered with $NO_3^-$ or $HSO_4^-$. In the positive mode, the ions are found as clusters with $NH_4^+$ or amines.

P. 10, l. 262 – 264: Are differences in condensation sink the only possible explanation for the "missing" nocturnal new particle formation in Hyytiälä? How do precursor concentrations compare? Have you done similar experiments in the CLOUD chamber, but with condensation sink? Please elaborate shortly on the meaning of similarity in observed spectra/ion composition between field and laboratory, but difference in new particle formation rates.

Indeed, there are also other factors explaining the fact that nocturnal events are not frequently observed in Hyytiälä. Using the data of CLOUD (Kirkby et al., 2016), which also includes this data, Gordon et al. (2016) did not find any nocturnal new particle formation at Hyytiälä in his model calculations. This is attributed mainly to the low terpene concentrations during night time. In addition, differences in condensation sink between this forested site and the CLOUD chamber may

also potentially contribute to the suppression of nocturnal nucleation. It has also to be considered that new particle formation rates smaller than 0.1 cm$^{-3}$s$^{-1}$ are hardly detected as such under ambient conditions, while they are measured in the CLOUD chamber. The concentrations of α-pinene used in this publication (258-618 ppt, see Table 1) are rather high for nocturnal concentrations in Hyytiälä (see Kontkanen et al., Atmos. Chem. Phys., 16, 13291–13307, 2016). What we show in Figure 2 is that the main HOMs components formed by α-pinene ozonolysis in the CLOUD chamber are also observed in Hyytiälä during night time. We do not consider here variations in relative intensities, which result from differences in i) concentrations; ii) temperature; iii) condensation sink. Presently, more experiments at the CLOUD chamber and more detailed analyses of measurements in Hyytiälä are underway to better understand the new particle formation mechanism. We therefore decided to delete this sentence and to set any speculation aside here as new particle formation is not addressed in this paper.

P. 11, l. 276 – 278: Do you have an explanation for this observation? Could differences in NH3+ mixing ratios for the different temperatures play a role?

NH$_3$ was present in the chamber as a contaminant and its level should not have changed over the course of the experiments. The gas phase concentration might be lower at lower temperatures as more NH$_3$ might condense to the walls. However, we did not see a change in the clustering of NH$_4^+$. Even at the lowest temperature, NH$_4^+$-clusters contributed more than 90% to the signal intensity. As written we observe the same features/trends for positive and negative mode spectra as seen from the C$_{20}$ band. However, in the positive mass spectrometer we can see a well separated C$_{40}$ band, which clearly shows the decrease in O/C with decreasing temperature. We would not expect a decrease in signal intensity of high O/C compounds if not due to a decrease in concentration because the clustering energy is highest for these. In the negative mass spectrometer the signal of the C$_{40}$ band is too noisy. This might have to do with lower sensitivity of the instrument at higher m/z or lower concentrations of C$_{40}$ clusters.

We changed the text as following: *Although the observations described in the following are valid for both polarities, the trends at the three temperatures are better seen in the positive mass spectra due to a higher sensitivity at high m/z.*

P. 11, l.279 – 280: This would be easier to see if y-axis ranges were expanded to beyond 0.2

We agree with the referee; however it was difficult to plot in a single Figure both polarities, at the three temperatures and show the signals (y-axis) of quite different intensities depending on the band. All the intensities are seen in Figure 4. Please find below another version of the plot with the y-axis beyond 0.2. In this presentation, the higher bands cannot be seen anymore. The full scale of intensities at 5°C can be seen in Figures 1B and 2. We refer to these now. We would like to keep the figure as it is.

[Figure]

P. 11, l. 279 – 288: Presumably these measurements are during nucleation/before the onset of particle growth, but can you rule out influence of condensation to the walls in this shift? And related, it is not well discernible from the figure that the rate of dimer formation at -25 °C is reduced, there could also simply be an increase in monomer formation rate. Please clarify.

Yes, the measurements were done before and during nucleation time. If wall loss rate influenced the concentration ratio of dimer to monomer, we should see the opposite trend. At lower temperature the monomers would stick better to the walls than at higher temperatures and their concentrations should decrease. As a matter of fact, most HOMs ($C_{10}$ and $C_{20}$) condense with the highest possible rate to the walls similarly as sulfuric acid. Thus, wall loss does not influence this shift.

Dimers are only a small fraction of the monomers. The reason, that the dimer signal is still larger than the monomer signal at higher temperatures is due to the higher cluster formation energies for dimers as seen in Table 3. The reviewer suggests that an increase of monomer formation could also

lead to this trend. The two processes are coupled. If less $RO_2$ radicals are formed that can combine to dimers, they end up as monomers. Thus, the monomer concentration increases while the dimer concentration decreases. Because dimers are only a small fraction of monomers the increase in monomer concentration is small. Thus, the strongest influence on the signal is caused by a decrease of dimers. Another reason for a shift in signal intensity could be a stronger stabilization of monomer $NH_4^+$ clusters at lower temperatures than of dimer clusters. Although the QC calculations given in Table 3 do not point in this direction, we cannot rule it out for other compounds than those used in the calculations.

P. 15, l. 368 - p. 16, l. 379: What exactly is meant by "better consistency"? Overlapping of ion masses? Can one expect ion rate closure? Y-axes labels would suggest otherwise, but it is hard to see.

Yes, we mean the better agreement between the simulated and measured mass spectrum of tetramers. We have reformulated this sentence to "*The better agreement of the modeled mass spectrum of the tetramer band in the later case with the measured one suggests that only the molecules with O/C ≥ 0.4 are able to form the tetramer cluster.*"
We do not understand the question on ion rate closure. We show here which combinations of dimers could lead to the tetramer pattern. We cannot predict the intensity distribution well, as this depends on the clustering energies of the $NH_4^+$ with the organic cluster. We assume here that it represents somewhat the actual concentration distribution of dimers and tetramers. This premise has been added now: "*One has to note that the comparison of modeled and measured spectrum relies on the assumption that the charge distribution of dimers is also reflected in the tetramers*".

P. 22, l. 513: Shouldn0 t the sentence finish with "with lower temperature"?

No. Ion clusters are more stable at all temperatures compared to neutral clusters. Therefore, ion clusters evaporate at lower rates and can grow faster. We slightly modified the sentence: *Finally, a comparison of the ΔG values as presented in Table 3 confirms the expected higher stability of charged clusters compared to neutral clusters, decreasing the evaporation rate of the nucleating clusters and enhancing new particle formation.*

P. 23, l. 544: You mention O:C ratio in a subclause. O:C ratios are being discussed as a determining factor in many processes related to atmospheric new particle number or mass formation. Your results imply temperature to be just as important. Please elaborate further on such an important implication.

Thank you for this comment. We added the following sentences: Nucleation and early growth is driven by the extremely low volatility compounds, i.e. dimers and monomers of high O:C ratios (Tröstl et al., 2016). Here, we observe a reduction of the autoxidation rate leading to oxidation products with lower O:C ratios with decreasing temperature. We expect that this is accompanied by a reduction of nucleation rates. However, a lower temperature reduces evaporation rates of clusters and thereby supports nucleation. The relative magnitude of these compensating effects will be subject of further investigations.

Technical corrections

P. 8, l. 221 No reference to Figure 1A. Figure 4 would profit from larger axis labels

We added a reference to Figure 1A. We agree with the referee and increased the size of the axis and labels as much as possible.

P. 14, l. 329: Dot is missing

This was corrected.

P. 22, l. 491: Delta is not printed properly

Thank you for the observation, this has been corrected

**References**

Bianchi et al. 2016 New particle formation in the free troposphere: A question of chemistry and timing. Science, 352 (6289), 1109–1112.

Ehn et al. 2010. Composition and temporal behavior of ambient ions in the boreal forest. Atmos. Chem. Phys., 10, 8513-8530

Frege et al. 2017. Chemical characterization of atmospheric ions at the high altitude research station Jungfraujoch (Switzerland). Atmos. Chem. Phys., 17, 2613-2629

Gordon et al. 2016. Reduced anthropogenic aerosol radiative forcing caused by biogenic new particle formation. Proceedings of the National Academy of Sciences, 113 (43). pp. 12053-12058.

Juninen et al. 2010. A high-resolution mass spectrometer to measure atmospheric ion composition. Atmos. Meas. Tech., 3, 1039–1053

Kirkby et al., 2016. Ion-induced nucleation of pure biogenic particles. Nature 533, 521–526

Kontkanen et al. 2016. Simple proxies for estimating the concentrations of monoterpenes and their oxidation products at a boreal forest site. Atmos. Chem. Phys., 16, 13291-13307

Kulmala et al. 2013. Direct Observations of Atmospheric Aerosol Nucleation. Science, 339, 943–946.

Suni et al. 2008. Formation and characteristics of ions and charged aerosol particles in a native Australian Eucalypt forest. Atmos. Chem. Phys., 8, 129-139.

Tröstl et al. 2016. The role of low-volatility organic compounds for initial particle growth in the atmosphere. Nature, 533, 527–531.

---

## Author Comment (AC2) · 17 Oct 2017

*"Influence of temperature on the molecular composition of ions and charged clusters during pure biogenic nucleation" by Carla Frege et al.*

**Anonymous Referee #2**

The manuscript describes laboratory experiments in the CERN CLOUD chamber. The authors investigate the molecular composition of positive and negative HOM clusters measured with APi-TOFs at three different temperatures (25∘C, 5∘C and -25∘C). The authors discuss the cluster formation in the positive and negative ion mode, the average oxygen-to-carbon (O:C) ratios focusing on the influence of the changing temperature. They essentially conclude a decrease in the rate of autoxidation with decreasing temperature. The experimental findings are supported by quantum chemical calculations of the binding energies of representative neutral and charged clusters. In general, the paper is well written and presents an interesting topic that is well suited to be published in ACP. The molecular processes of new particle formation, especially if organic molecules are also involved, are not well understood. Therefore, I suggest to publish the manuscript in ACP after considering the following comments.

Page 6, line 167: Impurities from alpha-pinene or chamber background ? Could that be signals from previous experiments (e.g. pinanediol ?)

No, there were no previous experiments with a different molecule. The impurities are most likely from evaporation of α-pinene since their signal increased with the injection of α-pinene to the chamber.

Page 7, line 197: What does "under relatively dry conditions" mean ?

Dry conditions mean basically RH = 0%. This has been clarified. Please also see the plot describing the behavior of the ions as a function of RH in the answer to Referee #1.

Page 10, line 257,258: The ion $C_{20}H_{32}O_{13}.NO_3-$ is mentioned two times ?

Thank you for the observation. This has been corrected.

Page 14, line 329: ". . . bands For a . . ." !?

There was a dot missing. This was corrected.

Page: 14, Table 2: The O/C-ratios from the monomers to the dimers are increasing in the positive mode and decreasing in the negative mode. While the decrease could be a result of oxygen loss in the formation of the dimers (in this case covalently bonded dimers (e.g. condensation reactions !?)), the increase in the O/C-ratio is more difficult to explain. One possibility would be the preferred formation from monomers with a rather high O/C-ratio, similar what the authors use as explanation for the tetramer formation, however, I wonder if the authors also included the C10H14OH+ signal in their signal weighted average O/C calculation (looking at figure 4 it seems so). In this case the inclusion of the background signal is of course misleading and should be corrected.

No, the signal of the "contaminant" peaks (i.e. $C_{10}H_{14}OH^+$ and $C_{20}H_{28}O_2H^+$) was not included in the O/C average calculation. We think, as the referee mentions, that the dimers are formed preferentially by the combination of monomers with a high O/C. Because the "contaminant" molecules (with low oxygen content and higher signal) were not included in the calculations, an increase in the O/C ration makes sense under this hypothesis.

Page 14, line 341: The authors often use the expression "cores". I suggest using simply "compounds" or "molecules".

When using the term core we want describe the neutral molecule or molecules contained in the cluster as opposed to the ion that provides the charge. We think it is more convenient than to describe it as the "neutral molecules of the cluster" or the "organic fraction of the cluster", etc. We would like to keep the notation as is.

Page 14, line 345: Again, I have the impression that the authors refer to the background signals mentioned earlier (compounds containing 1 oxygen atom) !? The explanation given on the next page (main oxidation products) is definitely not satisfying (in other words: pinonaldehyde is C10H16O2 and pinonic acid C10H16O3 – a factor of 2-3 higher in oxygen than needed).

We understand that this paragraph could be misunderstood because of the background signal $C_{10}H_{14}OH^+$ and $C_{20}H_{28}O_2H^+$, but those signals were excluded in all calculations. It should be remembered that although the monomer is generalized as molecules containing ten carbon atoms ($C_{10}$), the band also comprises molecules with slightly less carbon atoms (p.8 l. 208-210). This paragraph also makes reference to those molecules.  Signals identified with O/C of 1 are from $C_{10}H_{16}O$ (this could be α-pinene oxide, which has been observed by others), $C_8H_{12}O$, $C_9H_{14}O$, etc. These do not belong to the strong peaks. Monomers with two or more oxygen atoms can be attributed to the main oxidation products, i.e. pinonaldehyde, pinonic acid, etc. We write now:

Ions with O:C ratio less than 0.3 are probably from the main known oxidation products like pinonaldehyde, pinonic acid, etc., but also from minor products like pinene oxide and other compounds that have not been identified so far.